

**Response of winter fine particulate matter concentrations to**
**emission and meteorology changes in North China**
**Meng Gao[1,2], G. R. Carmichael[1,2], P. E. Saide[2,*], Zifeng Lu[3], Man Yu[1,2,**], D. G. Streets[3],**
**Zifa Wang[4]**
[1]Department of Chemical and Biochemical Engineering, University of Iowa, Iowa City, IA, USA
[2]Center for Global and Regional Environmental Research, University of Iowa, Iowa City, IA,
USA
[3]Energy Systems Division, Argonne National Laboratory, Argonne, IL, USA
[4] State Key Laboratory of Atmospheric Boundary Layer Physics and Atmospheric Chemistry,
Institute of Atmospheric Physics, Chinese Academy of Sciences, Beijing, China
[*]Now at Atmospheric Chemistry observations and Modeling (ACOM) lab, National Center for
Atmospheric Research (NCAR), Boulder, CO
[**]Now at Mathematics and Computer Science Division, Argonne National Laboratory, Argonne,
IL, USA
Correspondence to: M. Gao (meng-gao@uiowa.edu) and G. R. Carmichael
(gcarmich@engineering.uiowa.edu)



# 1  Abstract

The winter haze is a growing problem in North China, but the causes have not been well
understood. The chemistry version of the Weather Research and Forecasting model (WRF-
Chem) was applied in North China to examine how the $PM_{2.5}$ concentrations change in response
to changes in emissions (sulfur dioxide ($SO_2$), black carbon (BC), organic carbon (OC),
ammonia ($NH_3$), and nitrogen oxides ($NO_x$)), as well as meteorology (temperature, relative
humidity (RH), and wind speeds) changes in winter. From 1960 to 2010, the dramatic changes in
emissions lead to +260% increases in sulfate, +320% increases in nitrate, +300% increases in
ammonium, +160% increases in BC and 50% increases in OC. The responses of $PM_{2.5}$ to
individual emission specie indicate that the simultaneous increases in $SO_2$, $NH_3$ and $NO_x$
emissions dominated the increases in $PM_{2.5}$ concentrations.  $PM_{2.5}$ is more sensitive to changes in
$SO_2$ and $NH_3$ as compared to changes in $NO_x$ emissions. In addition, OC also accounts for a large
fraction in $PM_{2.5}$ changes. These results provide some implications for haze pollution control.
The responses of $PM_{2.5}$ concentrations to temperature increases are dominated by changes in
wind fields and mixing heights. $PM_{2.5}$ is not sensitive to temperature increases and RH decreases,
compared to changes in wind speed and aerosol feedbacks. From 1960 to 2010, aerosol
feedbacks have been significantly enhanced, due to higher aerosol loadings. The discussions in
this study indicate that dramatic changes in emissions are the main cause of increasing haze
events in North China, and long-term trends in atmospheric circulations maybe another
important cause since $PM_{2.5}$ is shown to be sensitive to wind speed and aerosol feedbacks. More
studies are necessary to get a better understanding of the aerosol-circulation interactions.



## 1 Introduction

PM$_{2.5}$ (particulate matter with diameter equal to or less than 2.5μm) is a main air pollution

concern due to its adverse effects on public health (Gao et al., 2015; Pope et al., 2009). Pope et

al. (2009) estimated that a decrease of 10μg PM$_{2.5}$ is related to about 0.6 year mean life

expectancy increase. PM$_{2.5}$ is also associated with visibility reduction and regional climate

(Cheung et al., 2005). Many cities in North China are experiencing severe haze pollution with

exceedingly high PM$_{2.5}$ concentrations. In January 2010, a regional haze occurred in North China

and maximum hourly PM$_{2.5}$ concentration in Tianjin was over 400μg/m$^3$ (Zhao et al., 2013). In

January 2013, another unprecedented haze event happened, and the daily PM$_{2.5}$ concentrations in

some areas of Beijing and Shijiazhuang reached over 500μg/m$^3$ (L. T. Wang et al., 2014), and

instantaneous PM$_{2.5}$ concentration at some urban measurement sites were over 1000μg/m$^3$

(Zheng et al., 2015).

It is well known that particulate matter levels are strongly influenced by emissions and

meteorological conditions (Steiner et al., 2006). The PM in the atmosphere can be directly

emitted from sources like wildfires, combustion, wind-blown dust, and sea-salt, or formed from

emitted gases through secondary aerosol formation mechanisms. Meteorology affects PM levels

via changing emissions, chemical reactions, transport and deposition processes (Mu and Liao,

2014). For example, increasing wildfire emission in North America is mainly caused by warmer

temperatures and precipitation changes (Dawson et al., 2014), and increased temperature leads to

higher biogenic emissions, which are important precursors of secondary organic aerosols

(Dawson et al., 2014; Heald et al., 2008; Jacob and Winner, 2009). Increasing temperature also

increases sulfate concentration due to higher SO$_2$ oxidation rates (Aw and Kleeman, 2003;

Dawson et al., 2007) and semi-volatile aerosols may decrease due to evaporation under higher



temperature (Sheehan and Bowman, 2001; Dawson et al., 2007; Tsigaridis and Kanakidou,
2007). Higher relative humidity (RH) favors the formation of nitrate and increasing precipitation
decreases all PM species via wet scavenging (Dawson et al., 2007; Tai et al., 2010). Furthermore,
increasing clouds promote in-cloud sulfate production (Tai et al., 2010) and changes in wind
speed and mixing height determines the dilution of primary and secondary PM (Jimenez-
Guerrero et al., 2012; Megaritis et al., 2014; Pay et al., 2012).
With rapid economic and industrial developments, emissions in China have grown during the
past years. It is estimated that $NO_x$ emissions in China increased by 70% from 1995 to 2004
(Zhang et al., 2007), Black Carbon (BC) by ~50% from 2000 to 2010 (Lu et al., 2011), Organic
Carbon (OC) by ~30% from 2000 to 2010 (Lu et al., 2011), and $SO_2$ by ~60% from 2000 to 2006
(Lu et al., 2011). Apart from emission changes, it was observed that the winter is warming up in
China, especially in the northern part (Guo et al., 2013; Hu et al., 2003; Ren et al., 2012). In
addition, wind speed in North China has lowered (Shi et al., 2015; Wang et al., 2004) and RH
has decreased in China (Song et al., 2012; Wang et al., 2004).
Many studies have investigated the impacts of emission changes on aerosol formation
(Aksoyoglu et al., 2011; Andreani-Aksoyoglu et al., 2008; Megaritis et al., 2013; Tsimpidi et al.,
2012a; Tsimpidi et al., 2012b) and the effects of climate/meteorology changes on $PM_{2.5}$
concentrations (Dawson et al., 2007; Megaritis et al., 2013; Megaritis et al., 2014; Tagaris et al.,
2007; Tai et al., 2012a; Tai et al., 2012b) in Europe and in the United States. The haze pollution
is growing in China, especially in North China, but the causes of the growth are not well
understood. For haze pollution in China, it has been reported that aerosol feedbacks that change
radiation and temperature can worsen pollution (Gao et al., 2016; Petäjä et al., 2016; Xing et al.,
2015c; Zhang et al., 2015).  In addition, the connections between haze and meteorological





conditions have been established in many former studies (Fu et al., 2014; Jia et al., 2015; Leng et
al., 2015; C. Li et al., 2015; Wang and Chen, 2016; Yang et al., 2016; X. Y. Zhang et al., 2015;
Zhang et al., 2016). However, the roles of the large emission changes during the last 4 to 5
decades and the observed meteorology changes in North China are not known.
The main objective of this study is to investigate the responses of $PM_{2.5}$ and its major species to
changes in emissions, including $SO_2$, BC, OC, $NO_x$ and $NH_3$, and to temperature, RH and wind
speed changes in the North China region. Winter haze in North China has a large contribution
from secondary inorganic aerosols and secondary inorganic aerosols are influenced by emissions,
temperature and RH. The models used in previous studies referenced above are all offline
models, which are not capable of considering the feedbacks of changing meteorology on other
meteorological variables, and the impacts of aerosols on meteorology. However, as pointed by
Gao et al. (2016) and J. Wang et al. (2014) aerosol feedbacks should not be neglected when
modeling aerosol in China. In this study, we consider aerosol feedbacks using the fully online
coupled WRF-Chem model.
This paper is organized as follows. First, the WRF-Chem model, model settings and domain
settings are briefly described and then in the next section, emission changes from 1960 to 2010
and accordingly $PM_{2.5}$ changes are discussed. After that, the responses of $PM_{2.5}$ to changes in
each emission species are analyzed. At last, the impacts of temperature, RH and wind speed
changes on $PM_{2.5}$ are analyzed and discussed.



**2 Methodology**
**2.1 WRF-Chem model**
The WRF-Chem model is the chemistry version of the Weather Research and Forecasting model,
which is fully online coupled, allowing gases and aerosols simulations at the same time as
meteorology simulations. In this study, we used a configuration that includes direct and indirect
feedbacks. The gas phase mechanism used in this study is the Carbon Bond Mechanism version
Z (CBM-Z), which includes 67 species and 164 reactions (Zaveri and Peters, 1999; Zaveri et al.,
2008). The gas-particle partitioning module used is the MOSAIC module, which considers all
important aerosol components, such as sulfate, nitrate, ammonium, BC, and OC (Zaveri et al.,
2008). Eight size bins version of MOSAIC was used and the aerosol sizes ranged from 0.039μm
to 10μm. Wind-blown dust was modeled online using the AFWA scheme. Two nested domains
with 81km and 27km grid resolutions from outer to innermost were used. Inputs into the model
include meteorological boundary and initial conditions (BCs and ICs) from NCEP FNL $1°×1°$
data and chemical boundary and initial conditions from MOZART model simulations (Emmons
et al., 2010). The anthropogenic emission inventory used is the MACCity (MACC/CityZEN EU
projects) emissions dataset, which provides monthly CO, $NO_x$, $SO_2$, VOC, BC, OC, and $NH_3$
emissions from different sectors for years between 1960 and 2020 (Granier et al., 2011). We
compared the MACCity emission inventory for 2010 (Granier et al., 2011) with MIX emission
inventory for 2010 (M. Li et al., 2015) in the China region, and the magnitudes of emissions in
China from these two datasets are very close. For example, the $SO_2$ emissions in China in 2010
were estimated to be 28663 Gg in the MIX emission inventory, and were 26876.3 Gg in the
MACCity emission inventory. Simulations for evaluating roles of emission changes were
conducted using emissions for year 1960 and year 2010. Biogenic emissions were estimated



online using the MEGAN model (Guenther et al., 2006). The simulation period was January
2010 and five days in previous month were modeled as spin-up to overcome the influences of
initial conditions.
**2.2 Sensitivity experiments**
We explored the sensitivities of the winter time haze event in 2010 to changes in emissions and
meteorology features through a series of simulations using 1960 and 2010 emission baselines.
Specifically, the influence of emission changes of $SO_2$, BC, OC, $NH_3$, and $NO_x$, and meteorology
(temperature, RH and wind speeds) changes on $PM_{2.5}$ and its major species was evaluated using a
series of simulations. They are listed and explained in Table 1. All base simulations use
meteorology of January 2010. It was pointed out surface air temperature in North China
increased at the rate of 0.36 ℃ per decade (Guo et al., 2013), the linear trends coefficient of
relative humidity anomaly in North China is about -0.60% per decade (Wang et al., 2004), and
national mean wind speed decreased 16% in the recent 50 years (Wang et al., 2004). To estimate
the impacts of changes in temperature, RH and wind speed that happened in the past several
decades, we decreased temperature by 2 degrees, increased RH by 10%, and increased wind
speeds by 20%, to reflect conditions of early decades. The changes of $PM_{2.5}$ and its major
components due to perturbations in emissions and meteorology are analyzed for the North China
region. The North China region is defined using domain 2 in Gao et al. (2016) and the statistics
of changes are calculated within domain 2 for the January 2010 month.



**2.3 Model Verification**
The WRF-Chem model performance has been evaluated using multiple observations, including
surface meteorological, chemical and optical data, and satellite data in Gao et al. (2016). The
model was shown to capture the variations of surface temperature, RH, while wind speed was
slightly overestimated (Gao et al., 2016), which has been reported as a common problem of
current WRF-Chem model under low wind speed conditions. The Root Mean Square Error
(RMSE) of temperature were all less than 3.2K and RMSEs of RH varied from 6.4 to 11.1%.
The RMSE of wind speeds were below the proposed criteria (2m/s) (Emery et al., 2001) at the
Beijing, Tianjin and Baoding stations, but larger than that criteria at the Chengde station. The
time series of simulated surface $PM_{2.5}$, $NO_2$, and $SO_2$ showed good agreement with observations
as did simulated aerosol optical depth (AOD) (Gao et al., 2016). Mean Fractional Bias (MFB)
ranged from -21.8% to 0.4% and Mean Fractional Error (MFE) ranged from 26.3% to 50.7%
when comparing against $PM_{2.5}$ observations (Gao et al., 2016). In addition, the comparison
between model results and satellite found that the vertical distribution of aerosol and horizontal
distribution were captured well by the model (Gao et al., 2016). Compared with observed $PM_{2.5}$
composition, sulfate and OC were underestimated and nitrate was overestimated by the model
(Gao et al., 2016). The underestimation of sulfate may be due to underestimation of $SO_2$ gas
phase oxidation, errors in aqueous-phase chemistry, and/or missing heterogeneous sulfate
formation (Gao et al., 2016).



## 3 Results and Discussion

### 3.1 PM$_{2.5}$ sensitivity to emission changes from 1960 to 2010

The emission changes of SO$_2$, NO$_x$, NH$_3$, BC and OC and resulting impacts on PM$_{2.5}$ from 1960 to 2010 were examined based on the MACCity dataset for years 1960 and 2010. Figure 1(a-e) displays SO$_2$, NO$_x$, NH$_3$, BC and OC emissions for 1960 and Figure 1(f-j) shows the changes from 1960 to 2010. Populated regions of North China, such as urban Beijing, urban Tianjin, and urban Shijiazhuang, exhibit large emissions of SO$_2$, NO$_x$, NH$_3$, BC and OC in 1960. However, NH$_3$ emissions exhibit different spatial distribution patterns from SO$_2$, NO$_x$, BC and OC emissions, because NH$_3$ is mainly associated with agriculture while SO$_2$, NO$_x$, BC and OC are mainly related with industrial and residential activities. From 1960 to 2010, SO$_2$, NO$_x$, NH$_3$, BC and OC increased over the entire North China domain and markedly increased in the Jing-Jin-Ji city cluster. In general, the domain averaged surface NO$_x$ emissions in North China increased by ~590% from 1960 to 2010. The domain averaged surface NH$_3$ emissions in North China increased by ~390% from 1960 to 2010, but the most significant increases occurred not in the Jing-Jin-Ji city cluster, but in Inner Mongolia. Unlike NH$_3$ emissions, BC emissions increased the most in urban Beijing from 1960 to 2010. This is because residential sources are the biggest contributor to BC in winter (Li et al., 2016) and the population in urban Beijing sharply increased with rapid urbanization. From 1960 to 2010, the mean BC emissions in North China increased by ~153%. Similar to BC emissions, OC emissions increased substantially in the center of Beijing, and the domain averaged increasing ratio is about 52% from 1960 to 2010. The enhancements of SO$_2$, NO$_x$, NH$_3$, BC and OC emissions in North China are expected to result in substantial increase in regional PM$_{2.5}$ concentrations.



Figure 2 shows the simulated monthly mean concentrations of $PM_{2.5}$ and its major components
(sulfate, nitrate, ammonium, BC and OC) based on emissions for year 1960. As listed in Table 2,
the domain averaged concentrations of sulfate, nitrate, ammonium, BC, OC, and $PM_{2.5}$ are 1.9,
0.8, 0.8, 1.5, 4.6, and 19.2$\mu g/m^3$, respectively. For year 1960, $PM_{2.5}$ concentrations are mainly
dominated by sulfate, OC and natural dust. Figure 3 displays the changes of sulfate, nitrate,
ammonium, BC, OC, and $PM_{2.5}$ due to changes in $SO_2$, $NO_x$, BC and OC emissions from 1960 to
2010. The predicted monthly mean concentrations of $PM_{2.5}$ components and $PM_{2.5}$ increase
everywhere over the entire domain due to emission changes resulting from the rapid urbanization
and industrialization from 1960 to 2010 (Figure 3(a-f)). As listed in Table 2, the predicted
monthly domain mean sulfate increases the largest (5.0 $\mu g/m^3$), followed by nitrate (2.6 $\mu g/m^3$)
and OC (2.5 $\mu g/m^3$).
From 1960 to 2010, the predicted BC increased by ~157% and OC increased by ~54% due to
153% increase in BC emissions and 52% increase in OC emissions. The nearly linear response of
both BC and OC aerosols to their emissions is due to the omission of a secondary organic aerosol
formation in the chosen CBMZ/MOSAIC mechanism. Thus, both of them were treated as
primary aerosols in these simulations. Our previous analyses indicate that SOA contribution in
this time period was small (Gao et al., 2016). The domain mean $PM_{2.5}$ concentrations increased
by 14.7$\mu g/m^3$ and the domain maximum increase is about 45$\mu g/m^3$ (Figure 3(f) and Table 2).
To explore how emission changes can affect haze days, we calculated the number of haze days in
urban Beijing for the CTL and EMI_2010 cases, using daily mean thresholds of 35 and 75$\mu g/m^3$
(China National Ambient Air Quality Grade I and Grade II Standard, L. T. Wang et al., 2014). In
urban Beijing, there are 4 days when daily mean $PM_{2.5}$ concentrations are above 35$\mu g/m^3$, and 0
days with daily mean $PM_{2.5}$ concentrations above 75$\mu g/m^3$ for the CTL case. For the EMI_2010



case, these two numbers increase to 15 and 8, indicating that the large increases in emissions
over the past several decades have significantly affected haze occurrences in Beijing.
**3.2 Sensitivity to changes in individual emission species**
The results discussed above show that in the winter period, the concentrations of secondary
inorganic aerosols (sulfate, nitrate, and ammonium) has increased dramatically. Thus it is
important to explore how sensitive secondary inorganic aerosol is to perturbations in precursor
emissions. Three sensitivity simulations (change $SO_2$, $NH_3$ and $NH_3$ emissions separately) were
conducted to examine how changes in emissions of each species affect aerosol concentrations.
The predicted changes of $PM_{2.5}$ and major $PM_{2.5}$ components at the ground-level are shown in
Figure 4 and monthly domain mean aerosol changes are summarized in Table 3.
3.2.1 Changes in $SO_2$ emissions
Due to changes in $SO_2$ emissions from 1960 to 2010, domain averaged sulfate increased by
3.4µg/m$^3$ (178.3%), nitrate decreased by -0.3µg/m$^3$ (-32.3%), and ammonium increased by
0.2µg/m$^3$ (29.4%). $NH_3$ reacts with sulfuric acid particles preferentially to its equilibrium with
gaseous nitric acid; hence neutralization of sulfuric acid is a necessary precondition to significant
particle ammonium nitrate formation (Seinfeld and Pandis, 2006).  Free $NH_3$ reacts with
enhanced $H_2SO_4$ due to increasing $SO_2$. As a result, ammonium increases and less $HNO_3$ gas is
transferred to the aerosol phase, which is consistent with the responses to increasing $SO_2$
emissions in Kharol et al. (2013).





### 3.2.2 Changes in $NH_3$ emissions
As shown in Figure 4 and Table 3, changes in $NH_3$ emissions from 1960 to 2010 result in
significant increases in nitrate ($1.5\mu g/m^3$, +76.0%) and ammonium ($0.6\mu g/m^3$, +84.0%). The
domain mean changes of sulfate due to increase in $NH_3$ is close to zero (about $0.1\mu g/m^3$),
because sulfate formation is only indirectly associated with $NH_3$ availability (Tsimpidi et al.,
2007). The significant changes in nitrate and ammonium occurred in south Hebei, Shandong, and
Henan province, where anthropogenic $NO_x$ emissions are very high (Figure 1). Although $NH_3$
emissions substantially increased in Inner Mongolia (Figure 1), responses of nitrate and
ammonium are not significant there. The substantial increases of nitrate after $NH_3$ emission
increase indicate that $NH_3$ limits the $NH_3NO_3$ formation in the North China region in this period.
### 3.2.3 Changes in $NO_x$ emissions
After changing $NO_x$ emissions from 1960 to 2010 levels, domain mean surface $PM_{2.5}$ decreases
by about $0.2\mu g/m^3$, but the changes of individual $PM_{2.5}$ inorganic components vary. The increase
of $NO_x$ emissions cause $0.7\mu g/m^3$ (-39.1%) decrease in monthly domain mean sulfate and the
domain peak sulfate reduction is about $2.9\mu g/m^3$. The OH radical is critical in the sulfate
formation in the regions where $SO_2$ concentrations are high and there is a competition between
$NO_x$ and VOCs to react with OH (Tsimpidi et al. 2012b). When the VOCs/$NO_x$ concentration
ratio is close to 5.5:1, the OH reacts with $NO_x$ and VOCs at an equal rate (Seinfeld and Pandis,
2006). When the concentration ratio is lower than 5.5:1, the OH primarily reacts with $NO_x$, and



the region with this concentration ratio is called VOC-limited region. In VOC-limited regions, an
increase of $NO_x$ will cause a decrease of OH and ozone concentration. When the VOCs/$NO_x$
concentration ratio is higher than 5.5:1, the OH will preferentially react with VOCs, and the
region with this high ratio is called $NO_x$-limited region. In the $NO_x$-limited region, an increase of
$NO_x$ will increase OH and ozone concentrations. In the simulated winter month, biogenic
emissions are low and $NO_x$ emissions in North China are very high, leading to lower VOCs to
$NO_x$ ratios, and it can be considered as VOC-limited region. Fu et al. (2012) pointed out that
north East Asia is VOC-limited in January and urban areas of Beijing are VOC-limited in both
January and July. As a result, the large increases in $NO_x$ emissions from 1960 to 2010 result in a
47.9% decrease in daytime surface ozone concentration and 55.6% decrease in daytime surface
OH concentration, which are shown in Figure 5. Over the entire domain, ozone and OH decrease
due to $NO_x$ emission increases (Figure 5). Consequently, sulfate aerosol decrease over the entire
domain, as shown in Figure 4(i). Decreases in sulfate might also be related to changes in
thermodynamics of the ammonium-sulfate-nitrate system. Although OH decreases, nitrate still
rises ($0.6\mu g/m^3$, +76.0%) due to the increase in $NO_x$ emissions. The domain mean ammonium
decreases by about 5.1% ($-0.04\mu g/m^3$). The net effects of $NO_x$ emission increases bring about
$0.2\mu g/m^3$ decrease in monthly domain mean $PM_{2.5}$ concentration and the domain peak decrease is
about $1.1\mu g/m^3$ (Table 3).
3.2.4 Coupled changes in $SO_2$, $NH_3$ and $NO_x$ emissions
As shown above, increasing $SO_2$ emissions significantly increases $PM_{2.5}$ concentrations in the
North China region, increasing $NH_3$ emissions also increases $PM_{2.5}$ concentrations but to a lesser
extent, and increasing $NO_x$ emissions slightly decreases $PM_{2.5}$ concentrations. The effects of



coupled changes in $SO_2$, $NH_3$ and $NO_x$ emissions are not a simple addition of the effect of changing
them separately. As listed in Table 3, the monthly domain mean sulfate, nitrate, ammonium, and
$PM_{2.5}$ increases more than the effects of changing emissions separately.  Domain mean sulfate
increases by 5.0μg/m$^3$ (+264.0%), nitrate increases by 2.6 μg/m$^3$ (+322.5%), ammonium
increases by 2.3μg/m$^3$ (295.2%) and $PM_{2.5}$ increases by 9.9μg/m$^3$ due to coupled changes in $SO_2$,
$NH_3$ and $NO_x$ emissions from 1960 to 2010. The simultaneous increases in $SO_2$, $NH_3$ and $NO_x$
emissions promote dramatic increases of secondary inorganic aerosols in North China.
3.2.5 Changes in BC and OC emissions
Since BC and OC are treated as primary aerosols in the chosen CBMZ/MOSAIC mechanism,
changes in their emissions do not show any impact on other aerosol components. As listed in
Table 3, monthly domain mean $PM_{2.5}$ increases by 2.3μg/m$^3$ and 2.5μg/m$^3$ due to changes in their
emissions from 1960 to 2010, respectively.
**3.3 Effects of temperature increases**
The model used in this study is a fully online-coupled model, which simulates meteorological
variables and chemical variables together. Therefore, it is not possible to increase temperature
uniformly, as was done in previous studies using offline models (Dawson et al., 2007; Megartis
et al., 2013; Megartis et al., 2014). To examine the sensitivity of $PM_{2.5}$ to temperature change
(reflecting the winter warming trends), we decrease temperature by 2℃ in the initial and
boundary conditions to reflect conditions more like 1960. As a result, the monthly domain mean
surface temperature increases 2.0 ℃ (CTL-CTL_T2), but in a non-uniform manner. The spatial



distributions of monthly mean surface temperature and temperature changes are shown in Figure
6(a). The monthly mean surface temperature increases more along top left domain boundaries
and less over the Bohai sea. The influence of increasing temperature on biogenic emissions is
included using temperature-sensitive biogenic emission model MEGAN (Guenther et al., 2006).
Due to the perturbation in temperature as mentioned above, sulfate, nitrate, ammonium and
$PM_{2.5}$ are predicted to increase in most areas of the domain (Figure 7). Predicted monthly mean
sulfate increases by 0.06µg/m$^3$ (+3.1%), nitrate increases by 0.03µg/m$^3$ (+4.2%), and ammonium
increases by 0.02µg/m$^3$ (+2.8%). The increases of sulfate, nitrate and ammonium are mostly
attributed to the increasing OH radicals, as shown in Figure 6(b). After the temperature
perturbation, daytime OH increases by about 3.6% on domain average. It was found that higher
temperature increased volatilization of ammonium nitrate and partitioned it to the gas phase
(Megaritis et al., 2014), but it is not significant here due to the low temperature in winter. In
addition, the increase of sulfate, nitrate, and ammonium could be partially due to accelerated gas-
phase reaction rate at higher temperature (Dawson et al., 2007; Megaritis et al., 2014).
As shown in Figure 7 (d-e), the concentrations of primary aerosols (BC and OC) also increase
after the temperature perturbations. This is due to changes in other physical parameter, such as
wind direction, wind speed, and PBLHs, which are key factors in the diffusion of air pollutants.
Figure 6(c) shows that monthly PBLHs in most North China areas decrease after the temperature
perturbation, and PBLHs over the Bohai sea decrease the most, with monthly mean decrease
over 50 meters. The monthly domain average daytime PBLHs decrease about 2.3% due to
increasing temperature. Surface horizontal winds also change (Figure 6(d)), which directly affect
the distributions and magnitudes of $PM_{2.5}$ concentrations in North China along with PBLH
changes.





The responses of PM$_{2.5}$ concentrations to temperature perturbation are different from the
responses of sulfate, nitrate, ammonium, BC and OC (Figure 7), with decreases in northwestern
regions and increases in most areas of the North China Plain. This is because natural dust is
dominant in northwestern regions (as shown in Figure 2(f)), and the concentrations of natural
dust decrease under lower horizontal wind speeds (Figure 6(d)). The monthly PM$_{2.5}$
concentration decreases by 0.01μg/m$^3$ on domain average due to temperature perturbation.
Because of temperature increase, the numbers of haze days (defined using the daily mean
threshold 35 and 75μg/m$^3$) in urban Beijing do not change.
The discussions shown above are based on emission levels in 1960. The responses to
temperature perturbations were also investigated based on emission levels in 2010, and the
results are shown in Figure S1, S2 and Table 3. The spatial distributions of the changes are
similar to the results shown above, but with larger magnitudes. The domain mean PBL heights
decreases slightly more (-8.6 compared to -8.3 meters). The domain mean PM$_{2.5}$ concentrations
and PM$_{2.5}$ components exhibit larger increases in North China, although daytime OH
concentrations increases less ($2.6\times10^{-9}$ compared to $3.3\times10^{-9}$ ppmv), suggesting that the
responses of PM$_{2.5}$ concentrations are mostly due to changes in PBL heights and wind fields.
**3.4 Effects of RH decreases**
The RH was enhanced by 10% in model initial and boundary conditions to represent RH for the
previous decades. As a result, the simulated monthly mean RH decreases by 9.3% on domain
average. Due to RH perturbation, domain mean PM$_{2.5}$ concentration decreases by 0.7 μg/m$^3$. As
shown Figure 8(a), PM$_{2.5}$ concentrations decrease in the Jing-Jin-Ji region but increase in




southern areas of the domain. The ammonium nitrate formation equilibrium depends on RH (Tai
et al., 2010), so $HNO_3$ may be shifted to the gas phase under lower RH. In addition, the changes
in RH can also affect the wet deposition rate. The increases in southern areas of the domain are
mainly due to suppressed in-cloud scavenging, as the decreases in RH inhibit the formation of
clouds. As shown in Figure 8(b), liquid water path (LWP) decreases by 75.0%. As a result, the
in-cloud scavenging loss rate decreases. The changes of predicted aerosol optical depth at 600nm
are shown in Figure 8(c). In most regions, visibility decreases due to lower RH. Because of RH
decreases, the numbers of haze days (defined using the daily mean threshold 35 and $75\mu g/m^3$) in
urban Beijing do not change. The responses to RH perturbations were also investigated based on
emission levels in 2010, and the results are shown in Figure S3 and Table 3. The responses are
also similar to changes based on emission levels in 1960, but with larger magnitudes.
**3.5 Effects of wind speed decreases**
Simulations were also carried out when wind speeds were increased to estimate the wind speeds
for the previous decades. The predicted domain averaged monthly mean wind speed decreases by
about 0.7 m/s. As shown in Figure 9(a), the monthly mean near surface horizontal winds are
pronounced in mountainous areas (northwest areas of the domain) and relatively smaller in other
areas. Figure 9(b) shows the changes of wind speeds (CTL-WS20) due to model perturbations.
The predicted monthly mean $PM_{2.5}$ concentrations decrease by $2.3\mu g/m^3$ on domain average, but
the responses of $PM_{2.5}$ vary within the domain. As shown in Figure 9(c), $PM_{2.5}$ concentrations
decrease in the northwestern areas because of lower production of natural dust under lower
horizontal wind speeds. However, in most areas of the North China Plain, $PM_{2.5}$ concentrations
increase under lower wind speeds (Figure 9(c)). The domain peak increase is about $2.4\,\mu g/m^3$,



which is based on low predicted PM$_{2.5}$ concentrations using emissions for year 1960. If the
concentration in base case is higher, the responses will be enhanced. As shown in Figure 9(d),
the domain maximum increases in PM$_{2.5}$ increases from 2.4 to 9.4 μg/m$^3$. Because of wind speed
decreases, number of haze days that daily mean PM$_{2.5}$ concentrations are above 35 μg/m$^3$
increases by 1.
**3.6 Effects of changes in aerosol feedbacks**
As mentioned in Gao et al. (2016), high concentrations of aerosol enhance stability of boundary
layer and increase PM$_{2.5}$ concentrations. Due to dramatic changes in emissions from 1960 to
2010, the strength of aerosol feedbacks may also have changed. To quantify these changes, we
simulated four cases (i.e., CTL, CTL_NF, EMI2010, and EMI2010_NF). CTL-CLT_NF and
EMI2010-EMI2010_NF are used to represent the contributions of aerosol radiative effects in
1960 and 2010. The changes in monthly mean daytime PBL heights and PM$_{2.5}$ concentrations are
shown in Figure 10. In 1960, the domain averaged PBL height decreases by 6.7 meters due to
aerosol radiative effects, and the domain maximum decrease is 25.4 meters. Correspondingly, the
domain averaged PM$_{2.5}$ increases by 0.1 μg/m$^3$ and the domain maximum increase is 0.9 μg/m$^3$. In
2010, the domain averaged PBL height decreases by 13.8 meters and the domain maximum
decrease is 55.2 meters (more than two times compared to 1960). Correspondingly, the domain
averaged PM$_{2.5}$ increases by 0.7 μg/m$^3$ and the domain maximum increase is 5.1 μg/m$^3$. The
enhanced strength of aerosol feedbacks is another important cause of degraded aerosol pollution.
Thus, controlling emissions will have a co-benefit of reducing strength of aerosol feedbacks.





**3.6 Implications for the effects of emission and meteorology changes on PM$_{2.5}$**
**concentrations**
The simulated responses of PM$_{2.5}$ concentrations to emission changes and meteorology changes
presented here, along with the previous presented effects of aerosol feedbacks (Gao et al. 2016),
provide important implications for the causes of the dramatic increases in winter PM$_{2.5}$
concentrations.
We calculated domain maximum changes in PM$_{2.5}$ concentration averaged over four stagnant
days (January 16-19) owing to emission changes from 1960-2010 (EMI2010-CTL), temperature
increases (CTL-CTL_T2), RH decreases (CTL-CTL_RH10), wind speed decreases (CTL-
CTL_RH20), and aerosol feedbacks (CTL-CTL_NF). The values are 137.7, 2.0, 2.6, 7.5 and
4.0μg/m$^3$, respectively. When the perturbations are based on emission levels in 2010, domain
maximum changes in PM$_{2.5}$ concentration due to temperature increases (EMI2010-
EMI2010_T2), RH decreases (EMI2010-EMI2010_RH10), wind speed decreases (EMI2010-
EMI2010_WS20), and aerosol feedbacks (EMI2010-EMI2010_NF) are 4.8, 4.7, 26.4 and
25.5μg/m$^3$. The effects of emission changes on haze formation are dominant and the effects of
aerosol feedbacks are comparable to the effects of wind speed decreases.
The comprehensive comparisons of these factors are also summarized in Table 3. Based on the
monthly domain mean responses of PM$_{2.5}$ concentrations to these factors, dramatic emission
changes due to urbanization and industrialization are the main causes of degraded air quality and
frequent haze occurrences in in North China. PM$_{2.5}$ is more sensitive to changes in SO$_2$, NH$_3$,
NO$_x$ emissions than BC and OC (about 106.3% higher). In addition, PM$_{2.5}$ is more sensitive to
changes in SO$_2$ and NH$_3$ emissions, as compared to changes in NO$_x$ emissions. Thus, they should
be preferentially controlled in order to reduce PM$_{2.5}$ levels. To control SO$_2$ emissions, the usage





of natural gas or other clean energy should be promoted to reduce the usage of coal. $NH_3$
emissions in China are mainly from agriculture sources (about 90%), including livestock,
fertilizer, and agricultural soil (Huang et al., 2012). Lelieveld et al. (2015) found that agricultural
emissions make the largest relative role in $PM_{2.5}$ concentration in eastern USA, Europe, Russia
and East Asia. To control $NH_3$ emissions from agriculture sources, some animal feeding and
animal housing strategies should be taken. In addition, controlling emissions will also have a co-
benefit of reducing strength of aerosol feedbacks.
According to the ECLIPSE_GAINS_4a emission dataset, $SO_2$ emissions in China will decrease
by -26%, $NO_x$ emissions in China will increase by 19%, and $NH_3$ emissions in China will
increase by 14% from 2010 to 2030. We predicted that these changes will lead to large decreases
in winter sulfate (-2.3μg/m$^3$ on domain average). Nitrate will increase by 1.5μg/m$^3$ and
ammonium will slightly decrease (-0.05μg/m$^3$) on domain average. The net change of domain
averaged $PM_{2.5}$ concentration is not significant (-0.8μg/m$^3$), so more efforts are needed to control
these important gaseous precursors.
From the information listed in Table 3, the responses of $PM_{2.5}$ concentrations to temperature and
RH perturbations are not as sensitive as to wind speed perturbations. From Sect. 3.3, we also
found that the effects of temperature perturbation on $PM_{2.5}$ concentration are dominant by
changes in PBLH and wind fields. Previous studies have pointed out the occurrences of haze
events are highly associated with atmospheric circulation anomalies (Chen and Wang, 2015;
Zhang et al., 2016). Thus, changes in atmospheric circulations may be another important cause of
growing haze pollution, in addition to emission changes. Furthermore, aerosol can also change
atmospheric circulation, especially in severely polluted East Asia. Thus, controlling emission
may have co-benefits of mitigate aerosol effects on atmospheric circulation.



The effects of changing atmospheric circulations on winter haze pollution in China is beyond the
scope of this paper, but should be investigated in future studies.
**4 Conclusions**
A fully online coupled meteorological and chemical transport model, WRF-Chem was used to
study responses of winter $PM_{2.5}$ concentrations to changes in emissions of $SO_2$, BC, OC, $NH_3$,
and $NO_x$ and to meteorology (temperature, RH, and wind speeds) changes in the North China
region, where people are suffering from severe winter haze pollution.
The detailed historical emissions dataset MACCity for year 1960 and 2010 were used to evaluate
the impacts of changes in emissions of $SO_2$, BC, and OC. From 1960 to 2010, the dramatic
changes in emissions lead to +264.0% increases in sulfate, +322.5% increases in nitrate,
+295.2% increases in ammonium, +157.0% increases in BC and 54% increases in OC. The
domain mean $PM_{2.5}$ concentrations increase by 14.7μg/m³ and the domain maximum increase is
about 45μg/m³. The responses of $PM_{2.5}$ to individual emission species indicate that the
simultaneous increases in $SO_2$, $NH_3$ and $NO_x$ emissions dominated the increases in $PM_{2.5}$
concentrations. $PM_{2.5}$ is more sensitive to $SO_2$ and $NH_3$ emission changes. The increases in $NO_x$
emissions may decrease surface ozone concentration and surface OH radical concentrations,
because the North China region is VOC-limited in the winter. In addition, OC accounts for a
large fraction in $PM_{2.5}$ changes.
The sensitivities of $PM_{2.5}$ to emission changes of its precursors provide some implications for
haze pollution control. $SO_2$, $NH_3$ and OC should be preferentially controlled. In China, the
residential sector, particularly biofuel usage is the primary sources of OC (Lu et al., 2011). The



usage of natural gas or other clean energy should be promoted to reduce the usage of coal and
biofuel to reduce $SO_2$ and OC. To control $NH_3$ emissions from agriculture sources, some animal
feeding and animal housing strategies should be taken.
The effects of changes in winter time meteorology conditions were also studied. Emission
changes from 1960 to 2010 substantially increase numbers of haze days, but meteorology
perturbations do not show any significant impacts. The perturbations in temperature and RH do
change $PM_{2.5}$ concentrations, but the strength is not as significant as the effects of wind speed
and emission changes. The effects of temperature perturbation are dominated by the changes in
surface wind fields and PBLHs. The effect of aerosol feedbacks is comparable to the effect of
decreasing wind speeds and the strength of aerosol feedbacks significantly increased from 1960
to 2010.
The above discussions indicate that aerosol concentrations are mainly controlled by atmospheric
circulations, except emission changes. Thus, long-term trends in atmospheric circulations maybe
another important cause of winter haze events in North China. More studies are necessary to get
a better understanding of the aerosol-circulation interactions.
**Acknowledgments**
This work was supported in part by Grants from NASA Applied Science (NNX11AI52G) and
EPA STAR (RD-83503701) programs. We thank the ECCAD website for providing the
MACCity emission inventory. We also would like to thank Dr. Yafang Cheng for her





contributions to the development of emission processing model. Contact M. Gao (meng-
gao@uiowa.edu) or G.R. Carmichael (gcarmich@engineering.uiowa.edu) for data requests.

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





1 Table 1. Simulation cases and descriptions

| Cases | Descriptions |
|---|---|
| CTL | Base case, anthropogenic emissions are from MACCity dataset for year 1960 |
| EMI2010 | Anthropogenic emissions are from MACCity dataset for year 2010 |
| $SO_2$-2010 | Same as CTL case except $SO_2$ emissions are for year 2010 |
| $NH_3$-2010 | Same as CTL case except $NH_3$ emissions are for year 2010 |
| $NO_x$-2010 | Same as CTL case except $NO_x$ emissions are for year 2010 |
| CTL_T2 | Same as CTL case except temperature BCs and ICs are decreased by 2K |
| CTL_RH10 | Same as CTL case except RH BCs and ICs are increased by 10% |
| CTL_WS20 | Same as CTL case except wind speed BCs and ICs are increased by 20% |
| CTL_NF | Same as CTL case except aerosol-radiation interactions are excluded |
| EMI2010_T2 | Same as EMI2010 case except temperature BCs and ICs are decreased by 2K |
| EMI2010_T2 | Same as EMI2010 case except RH BCs and ICs are increased by 10% |
| EMI2010_WS20 | Same as EMI2010 case except wind speed BCs and ICs are increased by 20% |
| EMI2010_NF | Same as EMI2010 case except aerosol-radiation interactions are excluded |



1  Table 2. Monthly domain mean concentrations of $PM_{2.5}$ and its major components for year 1960, and

2  domain maximum and mean concentrations for changes from 1960 to 2010 due to emission changes

3  $(\mu g/m^3)$

| Years | | $SO_4^{2-}$ | $NO_3^-$ | $NH_4^+$ | BC | OC | $PM_{2.5}$ |
|---|---|---|---|---|---|---|---|
| 1960 | Domain mean | 1.9 | 0.8 | 0.8 | 1.5 | 4.6 | 19.2 |
| 1960-2010 | Domain maximum | 18.9 | 7.8 | 6.8 | 9.9 | 11.1 | 45.0 |
| | Domain mean | 5.0 (264.0%) | 2.6 (322.5%) | 2.3 (295.2%) | 2.3 (156.6%) | 2.5 (54.0%) | 14.7 (76.4%) |



Table 3. Monthly domain mean changes of sulfate, nitrate, ammonium and $PM_{2.5}$ concentrations ($\mu g/m^3$)
due to emission and meteorology perturbations, and aerosol feedbacks (the two values of $PM_{2.5}$ changes
are for meteorology perturbations and aerosol feedbacks based on 1960 and 2010 emission levels,
respectively)

|  | $SO4^{2-}$ | $NO^{3-}$ | $NH^{4+}$ | $PM_{2.5}$ |
|---|---|---|---|---|
| Changes in $SO_2$ emissions | 3.4(178.3%) | -0.3 (-32.3%) | 0.2 (29.4%) | 3.4 |
| Changes in $NH_3$ emissions | 0.1 (5.3%) | 1.5 (189.6%) | 0.6 (84.0%) | 2.3 |
| Changes in $NO_x$ emissions | -0.7 (-39.1%) | 0.6 (76.0%) | -0.04 (-5.1%) | -0.2 |
| Changes in $SO_2$, $NH_3$, $NO_x$ emissions | 5.0 (264.0%) | 2.6 (322.5%) | 2.3 (295.2%) | 9.9 |
| Changes in BC emissions | - | - | - | 2.3 |
| Changes in OC emissions | - | - | - | 2.5 |
| Temperature perturbations | - | - | - | -0.01/0.3 |
| RH perturbations | - | - | - | -0.7/-1.1 |
| Wind speed perturbations | - | - | - | -2.3/-0.5 |
| Aerosol feedbacks |  |  |  | 0.1/0.7 |



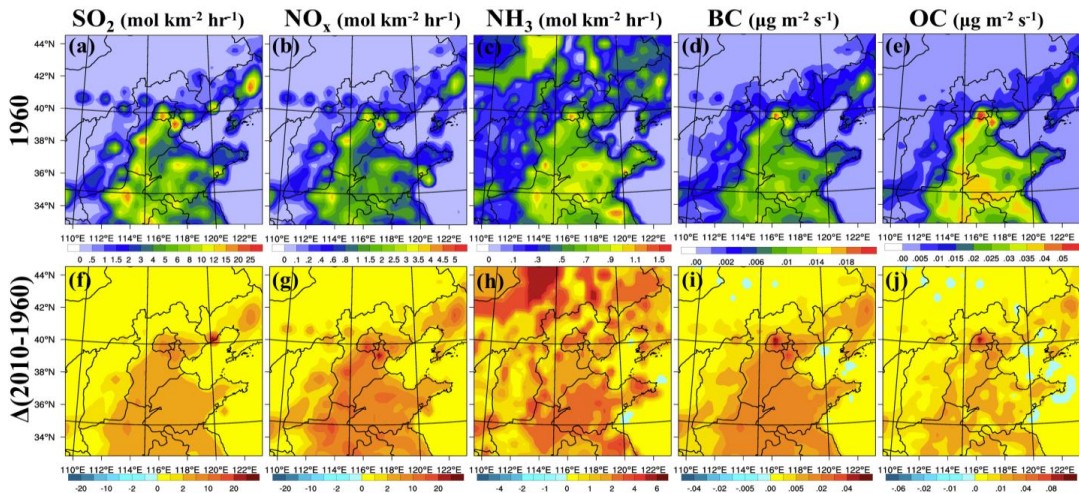

Figure 1. Surface $SO_2$, $NO_x$, $NH_3$, BC and OC emissions for year 1960 (a-e), and the changes of them
from 1960 to 2010 (f-j)

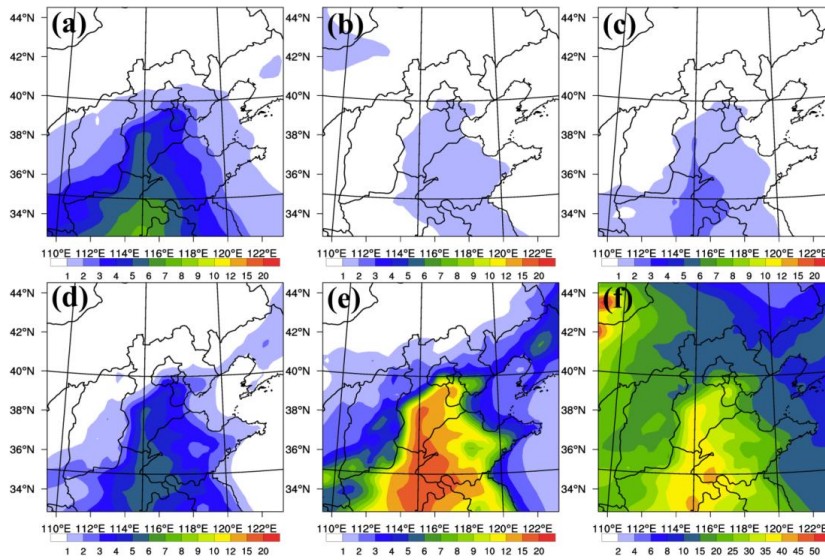

Figure 2. Predicted monthly mean sulfate (a), nitrate (b), ammonium (c), BC (d), OC (e) and $PM_{2.5}$ (f)
concentrations based on emissions for year 1960





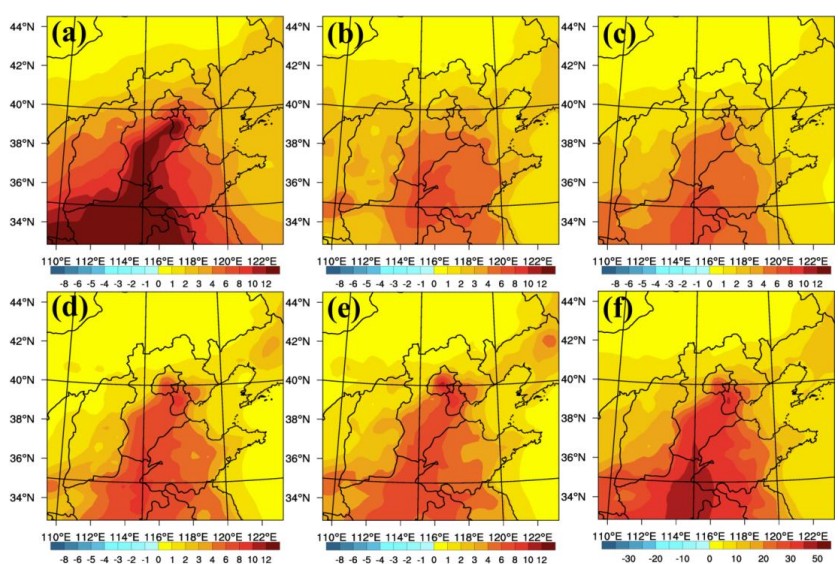

2          Figure 3. Predicted monthly mean changes of sulfate (a), nitrate (b), ammonium (c), BC (d), OC (e)

3                  and PM$_{2.5}$ (f) due to emission changes from 1960 to 2010 (units: μg/m$^3$)



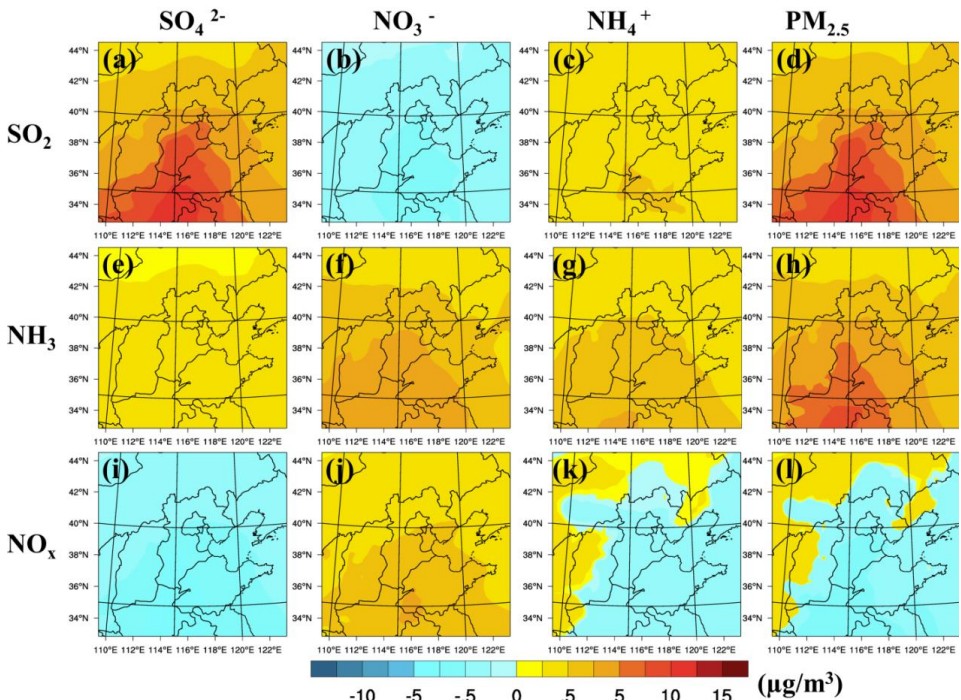

2   Figure 4. Responses of PM$_{2.5}$ and major PM$_{2.5}$ inorganic species (sulfate, nitrate, and ammonium) to

3   individual changes in SO$_2$, NH$_3$ and NO$_x$ emissions from 1960 to 2010

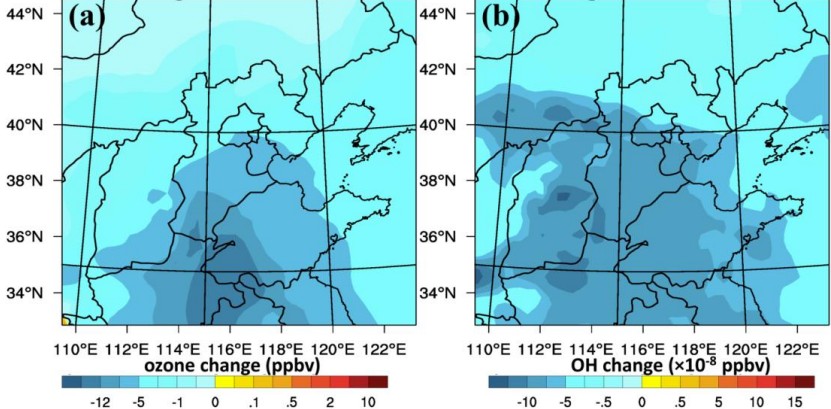

6   Figure 5. Daytime ozone (a) and daytime OH (b) changes due to NO$_x$ emission increases



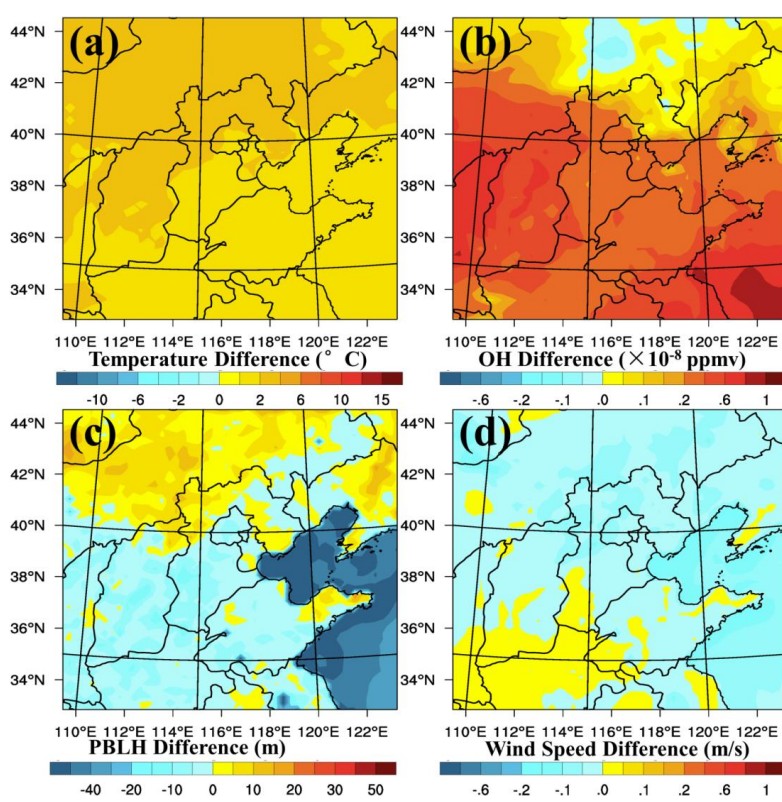

3    Figure 6. Monthly mean temperature difference due to perturbation in initial and boundary conditions

4    (a), and daily mean OH (b), mean PBLH (c) and mean near surface wind speed changes (d) due to

5                    temperature increase





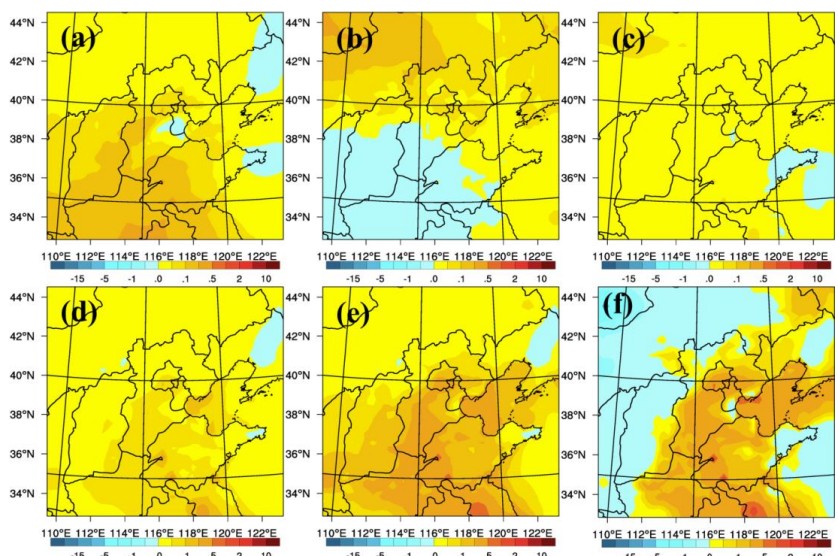

Figure 7. Monthly mean changes of sulfate (a), nitrate (b), ammonium (c), BC (d), OC (e), and PM$_{2.5}$

(f) and due to temperature increase

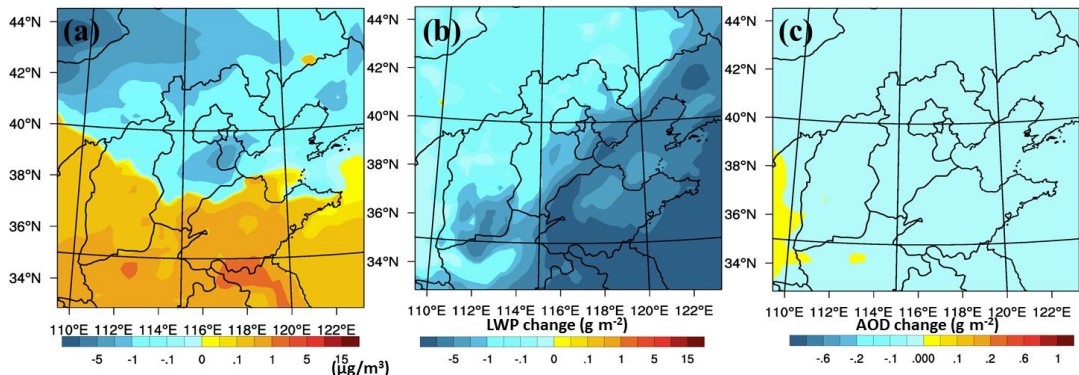

Figure 8. Monthly mean changes of PM$_{2.5}$ (a), LWP (b), and AOD at 600nm (c) due to RH decrease





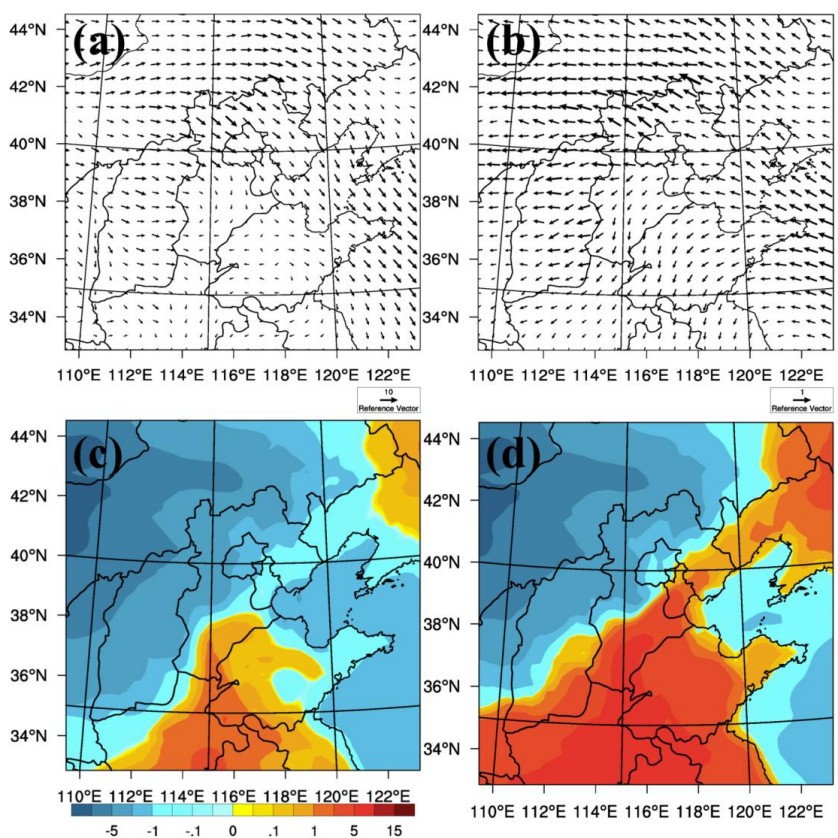

2          Figure 9. Monthly mean wind fields for WS20 case (a) and changes of wind speeds (CTL-

3          CTL_WS20) (b), and mean changes of $PM_{2.5}$ concentrations based on 1960 emission levels (c) and 2010

4                                       emission levels (d)



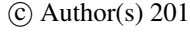

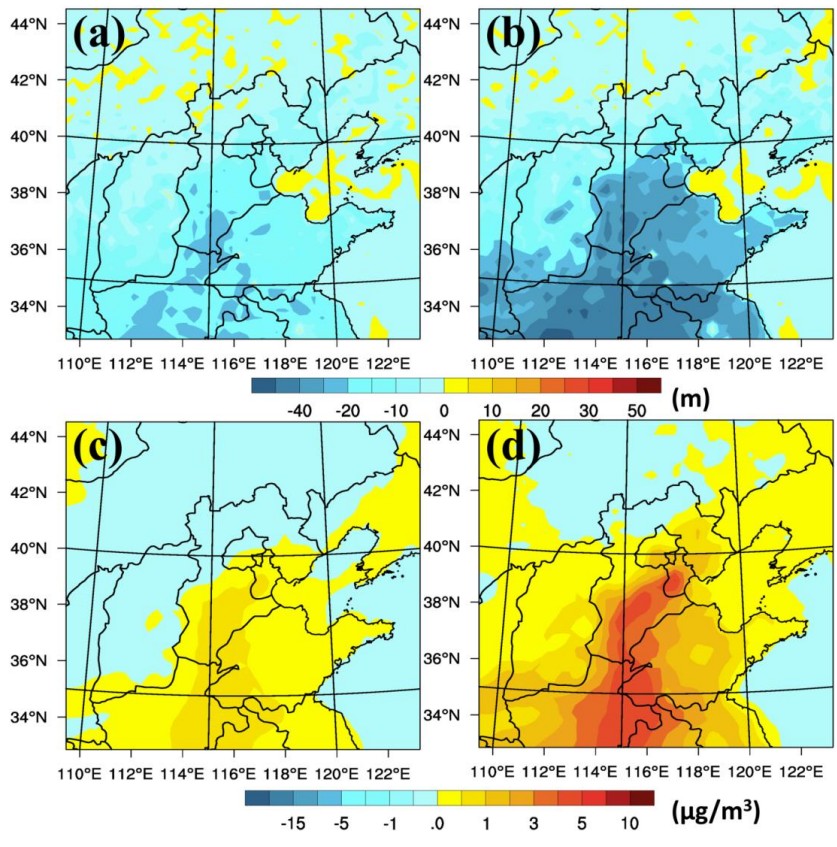

2    Figure 10. Monthly mean changes of daytime PBL heights for year 1960 (a) and 2010 (b), and of

3    daytime PM$_{2.5}$ concentrations for year 1960 (c) and 2010 (d) due to aerosol-radiation interactions

