# Peer review of "Response of winter fine particulate matter concentrations to emission and meteorology changes in North China"

_Atmospheric Chemistry and Physics, 2016_

## Referee Comment (RC1) · Anonymous Referee #1 · 2 Aug 2016

This paper investigated the responses of PM2.5 concentrations to changes in emissions (incl. SO2, NOx, NH3, BC and OC) and meteorology (incl. T, RH and wind speed) for a severe haze season in North China. Multiple sensitivity cases along with the baseline scenario in January 2010 were conducted with an online coupled meteorological and chemical transport model (i.e., WRF-Chem). The result suggests that dramatic changes in emissions are main cause of the increasing haze event in north China, and the winter PM2.5 is largely contributed by SO2 and NH3 emissions, as well as wind speed and aerosol feedbacks. The study is an important and very worthwhile exercise. Publication of the manuscript is recommended with minor revisions as suggested.

[Figure]

General Comments:

Detailed WRF configuration and emission processing is necessary. For example, there is little information regarding the WRF parameterization, nudging, how the feedback was set up. It is also necessary to provide basic details of model domain, particularly for the vertical profile. How the emissions were located vertically across the PBL? How the emissions and meteorological variables were adjusted in the sensitivity runs at different vertical heights?

Besides, discussion about the limitation of this study is necessary, such as lack of proper SOA simulation, no consideration of VOC and primary PM emission other than BC/OC. The simulation period was just one month in winter, results might be different in other seasons. The 13 simulation cases need to be discussed in the text, although they've been well summarized in Table 1. I would suggest to elaborate them in the method section.

Specific Comments:

Page 5 Line 9-10, some of the previous studies also used online coupled model, e.g., Wang et al ERL 2014, please clarify it.

Page 7 Line 16, did those changes apply to all vertical layers or just the ground? Is there any evidence to support those numbers?

Page 9 Line 12, elevated sources such as power plants and industry boilers are even more important than near-ground sources in China.

Page 10 Line 5, there is no data about natural dust provided in Table 2.

Page 10 Line 16-17, does that mean SOA is not important in China? It seems problematic.

Page 12 Line 10-11, please elaborate the reason.

Page 12 Line 11-12, NH3NO3 should be NH4NO3. The sentence is confusing. Does

that mean north China exhibits NH3 poor condition in winter?

Page 15 Line 20-21, please elaborate the reason, e.g., changes in T vertical profile?

Page 16 Line 8-9, a little confusing...but PM shows clear increases in Beijing.

Page 19 Line 20, the sensitivity represents the response per unit change, I suppose here it means the response to total changes. Please clarity it.

Page 20 Line 10, how was the prediction conducted?

Figure 1, I suggest to present total emissions in column rather than at surface only.

Figure 2, is it based on the meteorology in Jan 2010?

Figure 4, negative scale is too large to show any difference, please consider to modify it.

---

## Short Comment (SC1) · 16 Aug 2016

The authors perform emission and meteorology perturbation studies over East Asia (1960-2010), and I would like to point to the similarity of this paper to another paper in this special issue (Kasoar et al.).

I suggest that the magnitude of the emission perturbations Kasoar study (zero-out of anthropogenic) emissions, versus this study (1960-2010 emission changes) is sufficiently similar to warrant some comparison- e.g. of column amounts or AOD. Could something be said about 'local' temperature responses as calculated by the model?

One other analysis aspect that I find somewhat missing is not only the role of meteoro-

logical boundary conditions, but also the chemical ones. With some extra simulations (combinations of BC and emission perturbations) these aspects could also be evaluated, making the publication even more valuable for the HTAP special issue.

Below I offer some comments of technical nature that would warrant additional analysis.

1) Table 1: Assuming that PM2.5 in this study is defined as SO4+NO3+NH4+BC+OC, it seems that PM2.5 value in the first row (1960) is not correct.

2) Table 2: (and text). I find the difference between the single perturbation studies and the combined one surprisingly high. I am wondering if in the combined Sox-NOx-NH3 perturbation study also VOC and CO were perturbed- which perhaps could explain the large difference?

Anyway the authors should comment on this, because of cause the response of photochemistry to NOx perturbations can be quite different depending on VOC emissions. One diagnostic analysis is budget analysis: emissions, budget, transport (in-out), lifetime would be very valuable to show.

3) page 6, clarify whether the Mozart simulations also used 1960 (2010) emissions. The use of January 2010 warrants some discussion on how representative or typical this month was for a longer climatic period. While even for aerosol with lifetimes of a few days a spin-up of 5 days is rather short, it is certainly not capturing the lifetime of ozone and other components that feedback through oxidants on chemistry. As the authors seem to find large non-linear effects, I think they should consider trying to do longer simulations, if possible.

4) a table with domain emissions in experiments would be useful. It is not clear how much SO2 was changing (p.9). Clarify what was done with VOC, CO. I assume that the BC/IC were not changing along with the sensitivity studies, but it should be clarified.

5) section 3.4 (and 3.3) I am a bit wondering about the consistency of changing RH and T separately- while obvious the parameters are closely connected. I think this warrants more discussion. Would changing absolute humidity make more sense? Would dynamics change when changing RH?

---

## Referee Comment (RC2) · Anonymous Referee #2 · 22 Aug 2016

Overall I'd rate this manuscript as minor revisions - some rewriting is required, but I have no major concerns, aside from the missing SOA in the regional model. This is a significant omission - not so big that I'd require the authors to repeat their simulations with this fixed, but they definitely need to explain its absence and how this is justified.

My largest concern: Page 10, lines 14-15. I find the omission of a secondary organic aerosol formation mechanism in the model a concern, given the large amount of work in the literature showing the importance of this process towards net aerosol production. The authors reference Gao et al 2016 mentioning its impact is small – this reasoning should be repeated here. It's potentially a significant omission, given the increases in the region's VOC emissions – what are the unique local conditions that allow the au-

thors to justify its omission? Given the availability of several different parameterizations for SOA formation in the literature, why did the authors not ("just") include one in their model?

Not a "concern" per se, but something worth pointing out to the readers and perhaps investigating a bit further. Three points, all related:

(1) Page 7, lines 14 through 17: Later in the discussion the authors mention the manner in which this was done, by perturbing the initial and boundary conditions of the individual meteorological components to create these changes. This should be mentioned here as well, in a single sentence. This methodology later seems to result in a response from the fully coupled model which counteracts the meteorological perturbations. Some discussion of the mechanisms by which these counteracting effects takes place would be warranted.

(2) Sections 3.3,3.4, 3.5. The impact of these changes in the meteorology initial and boundary conditions may themselves be partially due to the response of the aerosols through feedbacks; affecting radiative transfer, etc. When the IC and BC temperature increases, the domain temperature decreases (section 3.3). When the IC and BC RH increases, the domain RH decreases (section 3.4). When the IC and BC wind speeds increase, the domain wind speeds decrease (question: section 3.5: I assume that the first sentence should read "carried out when initial and boundary condition wind speeds were increased"? Please explain in more detail how the winds were perturbed, and whether this was surface or 3D winds.

(3) Each of these meteorological perturbations to the initial and boundary conditions resulted in a model response which acted counter to that change. Would the authors concur that the feedback meteorological system is acting to damp or counteract meteorological perturbations? This is something worth mentioning in the paper, along with how the feedbacks act this way (e.g. temperature increases leading to increases in the type of aerosols which reduce surface temperatures, etc.).

Relatively minor issues:

Page 2, lines 19-20. The statement regarding long-term trends in atmospheric circulation potentially being important due to PM2.5's sensitivity to wind speed and aerosol feedbacks is unclear; it's not clear how the latter imply the former. I'm not sure, having scanned through the paper, that the authors have made a good case for that connection.

There are frequent references to Gao et al 2016 (perhaps intended as a companion paper) – for the benefit of those who do not have this paper, things like "domain 2 in Gao et al, 2016) (page 7 line 19) should be given more explicitly in this work (e.g. by showing the region on one of the figures). Similarly (Page 9, line 22), readers unfamiliar with the geography of the region might benefit from some symbols with the locations of the cities and regions mentioned appearing on the maps (maps are too small for names appearing on the maps themselves)

Page 3, line 22: would be better as "increases sulfate concentration due to the temperature dependence of SO2 oxidation and resulting higher SO2 oxidation rates".

Page 6 line 16: Please describe how the VOC emissions are speciated into CBM-Z VOCs in this description. That is, a speciation profiles must have been used – are these specific for different emissions sources, more generic, etc.? Or are the emissions data used already pre-speciated into the individual VOCs required for the model's chemical mechanism?

Page 8 line 19. There is recent work by McLinden et al in Nature Geoscience (May 2016), which uses satellite-based estimates of SO2 emissions to show regions and particular large sources which have been underreported in emissions inventories. Do the regions this reference shows have underestimates in SO2 production spatially correspond to the regions the authors of the current work have shown have underestimates in sulphate? If so, this would be worth mentioning.

[Figure]

Figure 3 discussion on page 10: to what extent do boundary conditions account for these changes? How much do the boundary conditions change between the two simulations? This should be discussed in the manuscript.

Page 12, line 18 to Page 13, line 11: Please include a plot of VOC:NOx ratios at the start and the end of the period to show how the ratio has changed in response to the emissions changes.

Page 15, lines 8-14: The OH increase has been attributed to the temperature perturbation (which makes sense in that this is the boundary condition which has been changed), but this does not necessarily mean that the temperature-dependent reaction rates are the main pathway by which temperature has increased the OH concentration. Another possible path might be through decreases in cloudiness leading to increases in photolysis, leading to increases in OH. Were there any changes in cloudiness in response to the temperature perturbation (or is this meteorological perturbation not fully interactive in which case, yes, temperature alone could be responsible for the OH change)?

Page 19, lines 21 to 23 versus Page 20, lines 10 through 12. In the first set of lines, the authors recommend reductions in SO2 and NH3 as a means of reduction particulate matter; in the second set of lines they show how increases in NH3 and NOx result in particle nitrate formation increasing in the future, despite SO2 decreases, in the winter. The authors need to clarify why / how NH3 is more important for future reductions of PM2.5 than NOx. Is the region relatively ammonia-poor, hence particle nitrate formation will be controlled by NH3 rather than NOx levels? The first set of statements needs to be justified, given the second set of statements, which could be due to either or both of the changes in NH3 and NOx.

---

## Author Comment (AC1) · 3 Sep 2016

**Response to referee comments on "Response of winter fine particulate matter concentrations to emission and meteorology changes in North China"**

We thank the reviewers for valuable comments. This document is organized as follows: the referees' comments are in black and our responses are in blue.

**To Referee #1**

This paper investigated the responses of $PM_{2.5}$ concentrations to changes in emissions (incl. $SO_2$, $NO_x$, $NH_3$, BC and OC) and meteorology (incl. T, RH and wind speed) for a severe haze season in North China. Multiple sensitivity cases along with the baseline scenario in January 2010 were conducted with an online coupled meteorological and chemical transport model (i.e., WRF-Chem). The result suggests that dramatic changes in emissions are main cause of the increasing haze event in north China, and the winter $PM_{2.5}$ is largely contributed by $SO_2$ and $NH_3$ emissions, as well as wind speed and aerosol feedbacks. The study is an important and very worthwhile exercise. Publication of the manuscript is recommended with minor revisions as suggested.

General Comments:

Detailed WRF configuration and emission processing is necessary. For example, there is little information regarding the WRF parameterization, nudging, how the feedback was set up. It is also necessary to provide basic details of model domain, particularly for the vertical profile. How the emissions were located vertically across the PBL? How the emissions and meteorological variables were adjusted in the sensitivity runs at different vertical heights?

Responses: Thanks for these great suggestions. We used same WRF parameterization as Gao et al., ACP, 2016 except the innermost 9km domain was not included here; The vertical profile used is the WRF default 27 vertical pressure profile; Analysis nudging was used for the outer domain; The feedback information was included in the method session: "In this study, we used a configuration that includes direct and indirect feedbacks."; The emissions were assigned to 6 layers from surface based on sources, for example emissions from large point sources (like chimneys) were assigned to higher layers; In the sensitivity runs, emissions and meteorological variables were uniformly adjusted at vertical layers.

We have added the following descriptions in the revised manuscript:

"Two nested domains with 81km and 27km horizontal grid resolutions from outer to innermost and 27 vertical grids were used (Figure S1 in supplementary material). Analysis nudging of meteorology variables was used for the outer domain. The model physics configurations also follow the settings in Gao et al. (2016)."

"We assigned emissions to the first 6 layers from surface based on sectors. For example, emissions from large point sources (such as chimneys) were assigned to higher layers."

"At different vertical heights, emission and meteorological variables were uniformly perturbed."

Besides, discussion about the limitation of this study is necessary, such as lack of proper SOA simulation, no consideration of VOC and primary PM emission other than BC/OC.

Responses: Thanks for these valuable suggestions.

About the lack of proper SOA simulation, we used the MADE/SORGRAM SOA scheme to investigate SOA during the same period in Gao et al, ACP, 2016 and found that it is not significant during winter haze, which might not be correct. However, current understanding of SOA is still limited and current SOA schemes largely underestimated SOA, especially during winter. Thus, we did not include it in this study.

Actually, VOC is considered in the model ("The anthropogenic emission inventory used is the MACCity (MACC/CityZEN EU projects) emissions dataset, which provides monthly CO, $NO_x$, $SO_2$, VOC, BC, OC, and $NH_3$ emissions from different sectors for years between 1960 and 2020).

We added the following paragraph to address limitations:

In our previous modeling study of the same period (January 2010), we found that SOA contribution was small, so we did not include SOA in this study. But this indication might be problematic due to current poorly parameterized SOA scheme. In the future, how changes in emissions and meteorology variables affect productions of SOA during winter should be further studied using more advanced SOA schemes. In addition, we did not consider primary PM except BC and OC in the model because there is no information in the MACCity emission inventory, which is another direction for improvements in future studies.

The simulation period was just one month in winter, results might be different in other seasons. The 13 simulation cases need to be discussed in the text, although they've been well summarized in Table 1. I would suggest to elaborate them in the method section.

Responses: We agree that the simulation in other seasons might be different, but it is beyond the scope of this paper. This paper focuses on winter haze pollution. We have elaborated the following descriptions of 13 simulation cases in method section.

"CTL case uses emissions for year 1960 and EMI2010 case uses emissions for year 2010. $SO_2$, $NH_3$, and $NO_x$ emissions were perturbed separately from 1960 to 2010 (i.e., $SO_2$-2010 $NH_3$-2010 $NO_x$-2010 cases). In the CTL_NF and EMI2010_NF cases, aerosol-radiation interactions are excluded based on emissions for year 1960 and 2010."

"reflect conditions of early decades (CTL_T2, CTL_RH10, CTL_WS20, EMI2010_T2, EMI2010_RH10, and EMI2010_WS20 cases)."

Specific Comments:

Page 5 Line 9-10, some of the previous studies also used online coupled model, e.g., Wang et al ERL 2014, please clarify it.

Responses: Thanks for this point. The focus of Wang et al ERL 2014 is aerosol feedbacks, while we were trying to say that previous studies about responses of $PM_{2.5}$ to changes in emissions and meteorology use offline models. Our expression (The models used in previous studies referenced above) might be confusing. So we changed the sentence to "The models used in previous studies of emission and meteorology perturbation referenced above". Hope it is clear now.

Page 7 Line 16, did those changes apply to all vertical layers or just the ground? Is there any evidence to support those numbers?

Responses: Thanks for this good question. These changes are uniformly applied to all vertical layers. The evidence are documented in those papers listed in line 11-14, Page 7. (i.e., It was pointed out surface air temperature in North China increased at the rate of 0.36 °C per decade (Guo et al., 2013), the linear trends coefficient of relative humidity anomaly in North China is about -0.60% per decade (Wang et al., 2004), and national mean wind speed decreased 16% in the recent 50 years (Wang et al., 2004)).

Page 9 Line 12, elevated sources such as power plants and industry boilers are even more important than near-ground sources in China.

Responses: We agree with this suggestion. We have plotted total column emissions and updated Figure 1 and the changing factors. We changed the sentence to "In general, the domain averaged $NO_x$ emissions in North China increased by ~990% from 1960 to 2010."

Page 10 Line 5, there is no data about natural dust provided in Table 2.

Responses: In Table 2, the sum of sulfate, nitrate, ammonium, OC and BC are smaller than $PM_{2.5}$. The differences mostly come from natural dust. Thanks for pointing this out. We have added this description in Page 10 Line 5 (natural dust (the difference between PM2.5 and the sum of sulfate, nitrate, ammonium, BC, OC) to make it clear.

Page 10 Line 16-17, does that mean SOA is not important in China? It seems problematic.

Responses: The simulated using MADE-SORGRAM shows that it is not important in winter in North China, which agrees with the previous SOA simulations in China (Jiang et al., 2012: Regional modeling of secondary organic aerosol over China using WRF/Chem). Currently, SOA is not well represented in the model due to incomplete understanding of SOA, so it might be problematic. We have added one paragraph in the summary session to mention this limitation.

Page 12 Line 10-11, please elaborate the reason.

Responses: We have added the reason: "due to trivial $NO_x$ emissions".

Page 12 Line 11-12, $NH_3NO_3$ should be $NH_4NO_3$. The sentence is confusing. Does that mean north China exhibits NH3 poor condition in winter?

Responses: Thanks for this correction. We have changed it to $NH_4NO_3$. That means $NH_3$ is relatively poor compared to $NO_x$. $NH_3$ also reacts with sulfuric acid to form $(NH_4)_2SO4$, so it may not be sufficient given the large amounts in $SO_2$ emissions. In addition, $NH_{3\ emission}$ is lower than in other seasons because $NH_3$ is mainly from agriculture and agriculture activity is reduced in winter.

Page 15 Line 20-21, please elaborate the reason, e.g., changes in T vertical profile?

Responses: Thanks for this great suggestion. We have added the reasons. "The monthly domain average daytime PBLHs decrease about 2.3% due to changes in temperature vertical profiles."

Page 16 Line 8-9, a little confusing...but PM shows clear increases in Beijing.

Responses: The amounts in changes are relatively small compared to PM concentrations in Beijing. We used daily mean threshold 35 and 75µg/m3 to define haze days. Due to T perturbation, the numbers of haze days do not show significant changes.

Page 19 Line 20, the sensitivity represents the response per unit change, I suppose here it means the response to total changes. Please clarity it.

Responses: Thanks for pointing this out. We have changed the expressions to "$PM_{2.5}$ shows more notable increases in response to changes in $SO_2$ and $NH_3$ as compared to increases in response to changes in $NO_x$ emissions". We also changed expressions of sensitive in other places.

Page 20 Line 10, how was the prediction conducted?

Responses: The predictions were conducted by perturbating emissions by those amounts. We have added the description ((by perturbating $SO_2$, $NO_x$ and $NH_3$ emissions by -26%, 19% and 14%)) to make it clear.

Figure 1, I suggest to present total emissions in column rather than at surface only.

Responses: Thanks for this suggestion. We have changed the Figure to total emissions in column.

Figure 2, is it based on the meteorology in Jan 2010?

Responses: Yes, it is based on the meteorology in Jan 2010.

Figure 4, negative scale is too large to show any difference, please consider to modify it.

Responses: Thanks for this good advice. We have modified the scale.

---

## Author Comment (AC2) · 6 Sep 2016

**Response to referee comments on "Response of winter fine particulate matter concentrations to emission and meteorology changes in North China"**

We thank the reviewers for valuable comments. This document is organized as follows: the referees' comments are in black and our responses are in blue.

**To Referee #2**

Overall I'd rate this manuscript as minor revisions - some rewriting is required, but I have no major concerns, aside from the missing SOA in the regional model. This is a significant omission - not so big that I'd require the authors to repeat their simulations with this fixed, but they definitely need to explain its absence and how this is justified.

My largest concern: Page 10, lines 14-15. I find the omission of a secondary organic aerosol formation mechanism in the model a concern, given the large amount of work in the literature showing the importance of this process towards net aerosol production. The authors reference Gao et al 2016 mentioning its impact is small – this reasoning should be repeated here. It's potentially a significant omission, given the increases in the region's VOC emissions – what are the unique local conditions that allow the authors to justify its omission? Given the availability of several different parameterizations for SOA formation in the literature, why did the authors not ("just") include one in their model?

Responses: Thanks for this important point. The simulated SOA using MADE-SORGRAM in my previous study showed that it is not important in winter in North China, which agrees with the previous SOA simulations in China (Jiang et al., 2012: Regional modeling of secondary organic aerosol over China using WRF/Chem). In Jiang et al. 2012, it was pointed out that high SOA in summer is due to high biogenic VOCs' emissions and intensive photochemical reactions (higher temperature and radiation in summer), and SOA is larger in South China. However, photochemical reaction is very weak in this study period due to high aerosol loadings (stagnant weather conditions in winter) and weak biogenic VOCs in winter. SOA is very complicated and not well understood, so the results from this SOA scheme might be problematic. Given current SOA modules is not good and the CBMZ-MOSAIC in WRF-Chem does not consider SOA, we did not include it in this study. We have added one paragraph in the summary session to mention this limitation.

"In our previous modeling study of the same period (January 2010), we found that SOA contribution was small, so we did not include SOA in this study. But this indication might be problematic due to current poorly parameterized SOA scheme. In the future, how changes in emissions and meteorology variables affect productions of SOA during winter should be further studied using more advanced SOA schemes. In addition, we did not consider primary PM except

BC and OC in the model because there is no information in the MACCity emission inventory, which is another direction for improvements in future studies."

Not a "concern" per se, but something worth pointing out to the readers and perhaps investigating a bit further. Three points, all related:

(1) Page 7, lines 14 through 17: Later in the discussion the authors mention the manner in which this was done, by perturbing the initial and boundary conditions of the individual meteorological components to create these changes. This should be mentioned here as well, in a single sentence. This methodology later seems to result in a response from the fully coupled model which counteracts the meteorological perturbations. Some discussion of the mechanisms by which these counteracting effects takes place would be warranted.

Responses: Thanks for this great point. We have added a single sentence description here. "These were conducted by perturbing the initial and boundary conditions of these individual meteorological variables".

For temperature and RH, the responses are generally consistent with perturbations (domain mean 2K increase for temperature and 9.3% decrease for RH). For wind speeds, we can see from Figure 9(a-b) that the monthly mean wind changes are in the opposite direction of the wind fields in base case, which is consistent with the decreasing perturbations. The responses of PM to decreases in wind speeds are also consistent, with increasing PM in highly polluted region and decreasing PM in northwest (when wind speeds are low, natural dust is low).

(2) Sections 3.3, 3.4, 3.5. The impact of these changes in the meteorology initial and boundary conditions may themselves be partially due to the response of the aerosols through feedbacks; affecting radiative transfer, etc. When the IC and BC temperature increases, the domain temperature decreases (section 3.3). When the IC and BC RH increases, the domain RH decreases (section 3.4). When the IC and BC wind speeds increase, the domain wind speeds decrease (question: section 3.5: I assume that the first sentence should read "carried out when initial and boundary condition wind speeds were increased"? Please explain in more detail how the winds were perturbed, and whether this was surface or 3D winds.

Responses: Thanks for this great question. It seems that the reviewer misunderstands our presentation here. We perturbed temperature, RH and wind speeds to represent them in 1960s.

In section 3.3, "we decrease temperature by 2 ℃ in the initial and boundary conditions to reflect conditions more like 1960. As a result, the monthly domain mean surface temperature increases 2.0 ℃". So, from 1960 to 2010, when IC and BC temperature increase, the domain temperature increases 2.0 ℃. We increases RH in IC and BC to represent 1960. So, from 1960 to 2010, IC and BC RH decreases, the domain RH decreases. For wind speeds, the case is similar. IC and BC wind speeds decrease from 1960 to 2010, and domain wind speeds decrease.

We agree that these changes are partially due to aerosol feedbacks, but the changing directions are generally consistent with the perturbations.

We have added "in initial and boundary conditions" to the first sentence of sect. 3.5. We added one sentence in sect. 2.2 to explain the perturbation in detail: "At different vertical heights, emission and meteorological variables were uniformly perturbed." The wind speeds were perturbed at different heights, not just surface.

(3) Each of these meteorological perturbations to the initial and boundary conditions resulted in a model response which acted counter to that change. Would the authors concur that the feedback meteorological system is acting to damp or counteract meteorological perturbations? This is something worth mentioning in the paper, along with how the feedbacks act this way (e.g. temperature increases leading to increases in the type of aerosols which reduce surface temperatures, etc.).

Responses: As I responded in point (1) and (2), the model responses are consistent with meteorological perturbations. We agree that feedbacks would partially affect meteorology, but it might not counteract meteorological perturbations. We have added sentences to mention this point in the revised manuscript.

Relatively minor issues:

Page 2, lines 19-20. The statement regarding long-term trends in atmospheric circulation potentially being important due to $PM_{2.5}$'s sensitivity to wind speed and aerosol feedbacks is unclear; it's not clear how the latter imply the former. I'm not sure, having scanned through the paper, that the authors have made a good case for that connection.

Responses: Since wind direction and speeds are mostly driven by position and intensity of large scale systems, such as Siberian High. Aerosol feedbacks change PM through suppressed PBL, while PBL height is also caused by the dominant large scale system (such as abnormal high at higher layers). This is just an indication for future studies.

There are frequent references to Gao et al 2016 (perhaps intended as a companion paper) – for the benefit of those who do not have this paper, things like "domain 2 in Gao et al, 2016) (page 7 line 19) should be given more explicitly in this work (e.g. by showing the region on one of the figures). Similarly (Page 9, line 22), readers unfamiliar with the geography of the region might benefit from some symbols with the locations of the cities and regions mentioned appearing on the maps (maps are too small for names appearing on the maps themselves)

Responses: This is really a good suggestion. We added map and description in the revised supplementary material for readers.

Page 3, line 22: would be better as "increases sulfate concentration due to the temperature dependence of SO2 oxidation and resulting higher SO2 oxidation rates".

Responses: Thanks for this suggestion. We have changed the sentence to this better form.

Page 6 line 16: Please describe how the VOC emissions are speciated into CBM-Z VOCs in this description. That is, a speciation profiles must have been used – are these specific for different emissions sources, more generic, etc.? Or are the emissions data used already pre-speciated into the individual VOCs required for the model's chemical mechanism?

Responses: The VOC emissions were speciated into CBMZ VOCs by referring the mapping information in (Meng Li et al., ACP, 2014, Mapping Asian anthropogenic emissions of non-methane volatile organic compounds to multiple chemical mechanisms) and the definitions of each species in the inventory and CBM-Z scheme. We use a generic distribution for different sources since detailed information is limited in the inventory.

Page 8 line 19. There is recent work by McLinden et al in Nature Geoscience (May 2016), which uses satellite-based estimates of SO2 emissions to show regions and particular large sources which have been underreported in emissions inventories. Do the regions this reference shows have underestimates in SO2 production spatially correspond to the regions the authors of the current work have shown have underestimates in sulphate? If so, this would be worth mentioning.

Responses: Thanks for mentioning this interesting new paper. In fact, we used surface $SO_2$ measurements to evaluate model and found $SO_2$ emissions might have been overestimated in the study region. In addition, we doubled $SO_2$ emissions and found it does not significantly increase sulphate. Thus, we don't think errors in $SO_2$ emission is the main cause.

Figure 3 discussion on page 10: to what extent do boundary conditions account for these changes? How much do the boundary conditions change between the two simulations? This should be discussed in the manuscript.

Responses: Results in Figure 3 are based on data in the innermost domain, which takes boundary conditions from the outer domain. To quantify the impact of changes in the outer domain, we simulated another case with the innermost domain emission fixed in 1960 and outer domain emission changed to 2010. The impacts of boundary conditions mostly occur around the south boundary, and show nearly no impact on PM2.5 in Beijing. We added one figure and discussion on Page 10.

"To quantify how much of the changes in Figure 3 are from the impacts of boundary conditions, we simulated another case with the innermost domain emissions fixed in 1960 and the outer domain emissions changed from 1960 to 2010. The impacts of boundary conditions mostly occur around the south boundary and show nearly no impact on $PM_{2.5}$ in Beijing (shown in Figure 4), which are consistent with the continuous weak southerly winds during the study period (Gao et al., 2016). On domain average, the impacts of boundary conditions result in 5.0ug/m$^3$ increase in the study domain, accounting for about 33.9% of the total changes in $PM_{2.5}$."

[Figure]

Figure 4 Changes of PM$_{2.5}$ due to boundary conditions from outer domain

Page 12, line 18 to Page 13, line 11: Please include a plot of VOC:NOx ratios at the start and the end of the period to show how the ratio has changed in response to the emissions changes.

Responses: Thanks for this great suggestion. We have added the plots of VOC:NOx ratios at the start and the end of the period. Before emission changes, the domain maximum VOCs: NOx ratio is 16.2 and domain mean ratio is 4.2. After emission changes, domain maximum ratio is 2.8 and domain averaged ratio is 1.2.

[Figure]

Figure 4 Averaged VOC:NOx ratios in 1960(a) and 2010(b)

Page 15, lines 8-14: The OH increase has been attributed to the temperature perturbation (which makes sense in that this is the boundary condition which has been changed), but this does not necessarily mean that the temperature-dependent reaction rates are the main pathway by which temperature has increased the OH concentration. Another possible path might be through decreases in cloudiness leading to increases in photolysis, leading to increases in OH. Were there any changes in cloudiness in response to the temperature perturbation (or is this meteorological perturbation not fully interactive in which case, yes, temperature alone could be responsible for the OH change)?

Responses: This is a good point that I did not consider. We have plotted the difference in liquid water path (which is the integration of cloud water) and added this factor into the revised manuscript.

[Figure]

Figure Changes in LWP due to temperature perturbations

Page 19, lines 21 to 23 versus Page 20, lines 10 through 12. In the first set of lines, the authors recommend reductions in SO2 and NH3 as a means of reduction particulate matter; in the second set of lines they show how increases in NH3 and NOx result in particle nitrate formation increasing in the future, despite SO2 decreases, in the winter. The authors need to clarify why / how NH3 is more important for future reductions of PM2.5 than NOx. Is the region relatively ammonia-poor, hence particle nitrate formation will be controlled by NH3 rather than NOx levels? The first set of statements needs to be justified, given the second set of statements, which could be due to either or both of the changes in NH3 and NOx.

Responses: Thanks for this advice for improvements. The reduction of NH3 is more important than NOx because (1) As shown in Figure 4, the increases in NH3 emissions lead to much more

PM than increases in NOx; In winter, this region is VOC limited, so the reduction in NOx might not be effective; (2) This region is relatively ammonia-poor in winter, which is consistent with previous findings in Europe (Megaritis et al., 2013) that reducing NH3 emissions seems to be the most effective control strategy to reduce $PM_{2.5}$. We have added justifications in the revised manuscript.

---

## Author Comment (AC3) · 6 Sep 2016

**Response to short comments on "Response of winter fine particulate matter concentrations to emission and meteorology changes in North China"**

We thank Dr. Frank Dentener for valuable comments. This document is organized as follows: the comments are in black and our responses are in blue.

**To Short Comments**

The authors perform emission and meteorology perturbation studies over East Asia (1960-2010), and I would like to point to the similarity of this paper to another paper in this special issue (Kasoar et al.).

I suggest that the magnitude of the emission perturbations Kasoar study (zero-out of anthropogenic) emissions, versus this study (1960-2010 emission changes) is sufficiently similar to warrant some comparison- e.g. of column amounts or AOD. Could something be said about 'local' temperature responses as calculated by the model? One other analysis aspect that I find somewhat missing is not only the role of meteorological boundary conditions, but also the chemical ones. With some extra simulations (combinations of BC and emission perturbations) these aspects could also be evaluated, making the publication even more valuable for the HTAP special issue.

Responses: Thanks for mentioning this great work (Kasoar et al.) published in the same issue. In Kasoar's study, the results are presented due to removal of SO2 emissions, but our results are due to all anthropogenic emission changes from 1960-2010. We plot column AOD and temperature responses and compared with the results in Kasoar's study. As shown below, due to reductions in 2010 from 2010 to 1960, AOD at 600nm decreases by -0.1~-0.4 in highly polluted areas, and temperature increases by about 0.1~0.55 degree. The results are closet to the results from HadGEM3-GA4 model in Kasoar study.

[Figure]

Figure Change in column AOD at 600nm (a) and near surface temperature (b)

Below I offer some comments of technical nature that would warrant additional analysis.

1) Table 1: Assuming that PM2.5 in this study is defined as SO4+NO3+NH4+BC+OC, it seems that PM2.5 value in the first row (1960) is not correct.

Responses: Thanks for this question. Actually, PM2.5 in this study is not just SO4+NO3+NH4+BC+OC. It also includes na,cl, and dust. Na and cl concentrations are very low, so we just ignore it in this study. In table 1, PM2.5 is larger than SO4+NO3+NH4+BC+OC because it includes dust. To clarify this, I added "natural dust (the difference between PM2.5 and the sum of sulfate, nitrate, ammonium, BC, OC)" in the revised manuscript. Hope it is not confusing now.

2) Table 2: (and text). I find the difference between the single perturbation studies and the combined one surprisingly high. I am wondering if in the combined Sox-NOx-NH3 perturbation study also VOC and CO were perturbed- which perhaps could explain the large difference?

Anyway the authors should comment on this, because of cause the response of photochemistry to NOx perturbations can be quite different depending on VOC emissions. One diagnostic analysis is budget analysis: emissions, budget, transport (in-out), lifetime would be very valuable to show.

Responses: Thanks for this great question and suggestion. Yes, you are right, in the combined Sox-NOx-NH3 perturbation study, VOC and CO were also perturbed, which might be the cause of the large difference. We have added this comment on the cause of this difference in the revised manuscript. We also added the plots of VOCs: NOx ratio in 1960 and 2010, which will help readers to understand the photochemical background in 1960 and 2010.

3) page 6, clarify whether the Mozart simulations also used 1960 (2010) emissions. The use of January 2010 warrants some discussion on how representative or typical this month was for a longer climatic period. While even for aerosol with lifetimes of a few days a spin-up of 5 days is rather short, it is certainly not capturing the lifetime of ozone and other components that feedback through oxidants on chemistry. As the authors seem to find large non-linear effects, I think they should consider trying to do longer simulations, if possible.

Responses: Thanks for pointing out this issue. The Mozart simulations used 2010 emissions but we only used boundary and initial conditions for the outer domain, which used emissions for 1960. We downloaded MOZART simulations from NCAR website, which does not cover early years (1960). The results presented used the innermost domain, which takes boundary conditions from outer domain. We have clarified this point in the revised manuscript. This study focuses on a month based simulation, but we are working on running long simulations from a climate change perspective, which will be another interesting story.

4) a table with domain emissions in experiments would be useful. It is not clear how much SO2 was changing (p.9). Clarify what was done with VOC, CO. I assume that the BC/IC were not changing along with the sensitivity studies, but it should be clarified.

Responses: Thanks for this great question. We have added the changing factor for $SO_2$ (increased by +220%) in the revised manuscript. Changes in other species were also added and modified based on column values. CO and VOCs were also projected from 1960 to 2010. We have added this clarification. Yes, IC/BCs were not changing. We have also added this point in the revised manuscript.

5) section 3.4 (and 3.3) I am a bit wondering about the consistency of changing RH and T separately- while obvious the parameters are closely connected. I think this warrants more discussion. Would changing absolute humidity make more sense? Would dynamics change when changing RH?

Responses: Thanks for this great question. Yes, T and RH are closely connected. The increases in T will lead to changes in RH, and the changed RH will affect some chemical reactions. But these changes are also due to changing T. Under global warming background, chemical reactions might change via changed RH. Some previous studies perturb absolute humidity using offline models. But it is not easy to implement it in the fully online coupled model. We found that perturbing initial and boundary conditions is one of the solutions, but in WRF, RH is provided in boundary and initial conditions, not absolute humidity. Changing absolute humidity might be more interesting and deserve future investigation. When changing RH, wind fields slightly changed, not significant.

---

## Editor Decision (ED1)

Minor revisions:

I would like to thank the authors for their detailed and thorough response to the two reviewer comments and the short comment by Dr. Frank Dentener and for revising the manuscript accordingly. Before accepting the manuscript for final publication, I would like to see the following minor issues addressed:

1) A discussion of the author's choice to use CBM-Z/MOSAIC which lacks a treatment of SOA should be included in section 2.1 "WRF-Chem Model". The authors do discuss the lack of SOA treatment - which was noted by both reviewers – in section 3.1 and in the summary, but it would be better to include this information up front.

2) Response to points (2) and (3) raised by reviewer #2: I have to admit that I was also initially confused by the description of these meteorological perturbation simulations and also thought that the authors were describing counteracting feedback effects. After rereading sections 3.3 – 3.5 a few more times, I realized that this potential misunderstanding is a wording issue that has not been resolved in the revised manuscript and needs to be addressed before final publication.

Take the following sentences from page 16: ".., we decrease temperature by 2C in the initial and boundary conditions to reflect conditions more like 1960. As a result, the monthly domain mean surface temperature increases 2C (CTL – CTL_T2), …". At first glance, these two sentences seem to suggest that decreasing temperatures in the IC/BC led to increases in simulated temperature, and this is the impression reviewer #2 and I both got. Upon re-reading the section, I realize what is meant here – temperatures were decreased from the 2010 met conditions used in CTL to represent 1960 as a sensitivity case, and consequently there is a temperature increase between CTL_T2 (representing 1960 met conditions) and CTL (representing 2010 met conditions). One way to clarify this might be to change these two sentences as follows ".., in the CTL_T2 simulation we decrease temperature by 2C in the initial and boundary conditions to reflect conditions more like those that occurred in 1960 rather than the 2010 conditions used in CTL. As a result of these changes, the monthly domain mean surface temperature increases 2C between CTL_T2 and CTL, …".

In a sense, the root cause of the confusion is that the authors treat 1960 as a reference point for their discussion of changes in emissions and meteorology, but the "CTL" experiment does not reflect that reference point but rather relies on 2010 meteorology. This potential for confusion runs throughout sections 3.3 – 3.5 whenever the authors discuss "perturbations" since this term actually seems to refer to the approximated 1960 – 2010 meteorological change rather than the actual perturbation simulation which was a reverse meteorological change meant to "revert" the 2010 meteorological conditions in CTL to 1960 conditions. Examples: page 16, lines 13-16, page 18, lines 8-11, page 19, lines 2-6. I suggest that the author consider using wording like "Due to the approximated change in temperature/Rh/winds between 1960 and 2010" rather than "Due to the perturbation" in these instances.

3) I assume that all emission and meteorological perturbations were only implemented in the innermost domain (except the boundary condition sensitivity simulation where emission were

perturbed in the outermost domain). If so, please clarify this in section 2.2. If not, please clarify how the analysis nudging applied in the outermost domain was applied for the meteorological perturbation cases.

4) Effect of chemical boundary conditions, page 11, lines 14 – 20. Please clarify that this test investigates changes due to boundary conditions only for the innermost domain, not the outermost domain. In other words, clarify that you are investigating the effects of regional emission changes in the 81 km domain on results in the 27 km domain and that your analysis does not account for the potential effects of changes in global atmospheric composition between 1960 and 2010 since the same 2010 MOZART simulation was used to derive boundary conditions for the outermost domain in all simulations. In addition, the boundary condition sensitivity simulation shown in Figure S2 should be added to Table 1.

5) Section 3.2.4. As noted by reviewer #2 and acknowledged by the authors in the new sentence added at the end of this section, a main reason for why the results of the SO2, NH3 and NOx emission sensitivity cases do not add up to the full difference between the CTL and EMI2010 cases is the change of other emissions such as VOC in the latter. Therefore, it is somewhat misleading to label this section "coupled changes in SO2, NH3 and NOx emissions" – a better choice might be "Comparison of individual changes in SO2, NH3 and NOx emissions to simultaneous changes in all emissions" and to state up front that the latter case includes changes in emissions besides SO2, NH3 and NOx.

6) As noted by reviewer #1, the PBL decreases associated with the 2C temperature increase are noteworthy. The authors responded that they agree with the reviewer's suggestion and revised the manuscript to state that changes in vertical temperature profiles are causing the decrease in daytime PBL ("the monthly domain average daytime PBLHs decrease about 2.3% due to changes in temperature vertical profiles"). Please add more information on your analysis of how the vertical temperature profiles changed as a result of the uniform 2C increase in temperature initial and boundary conditions. Or do potential changes in wind speed induced by these temperature changes also play a role?

7) The sensitivity simulation with the ECLIPSE_GAINS_4a emissions dataset discussed on pages 21/22 should be added to Table 1.

8) Section 3.6. If the authors have any thoughts on whether direct or indirect aerosol radiative effects are more important in driving these changes in PBL heights, it would be good to include this discussion here. This might include comparing changes in clear-sky and all-sky radiation and clouds. However, this is just a suggestion and not required since it is a relatively minor aspect of the overall study and may best be left for future work.

9) While the manuscript is generally well written, it would benefit from careful proofreading for language and grammar. A few specific examples are page 7 line 22 "(i.g. SO2-2010 NH3-2010 NOx-2010 cases)" which probably should be "(i.e. SO2-2010, NH3-2010, and NOx-2010)", page 8, line 1 "pointed out surface air temperature … increased" which should be "pointed out that

surface air temperature … increased", page 8, line 15 "for the January 2010 month" which should be "for the month of January 2010", and page 8 line 20 "variations of surface temperature, RH" which should be "variations of surface temperature and RH".